# Stability of Manganese(II)–Pyrazine, –Quinoxaline or –Phenazine Complexes and Their Potential as Carbonate Sequestration Agents

**DOI:** 10.3390/molecules27051648

**Published:** 2022-03-02

**Authors:** José J. N. Segoviano-Garfias, Gabriela A. Zanor, Fidel Ávila-Ramos, Egla Yareth Bivián-Castro

**Affiliations:** 1División de Ciencias de la Vida (DICIVA), Universidad de Guanajuato, Campus Irapuato-Salamanca, Ex Hacienda El Copal, Carretera Irapuato-Silao Km. 9, Irapuato 36500, Mexico; gzanor@ugto.mx (G.A.Z.); ledifar@ugto.mx (F.Á.-R.); 2Centro Universitario de los Lagos, Universidad de Guadalajara, Enrique Díaz de León 1144, Col. Paseos de la Montaña, Lagos de Moreno 47460, Mexico; egla.bivian@academicos.udg.mx

**Keywords:** carbonate capture, manganese(II) complexes, formation constants

## Abstract

Carbonate sequestration technology is a complement of CO_2_ sequestration technology, which might assure its long-term viability. In this work, in order to explore the interactions between Mn^2+^ ion with several ligands and carbonate ion, we reported a spectrophotometric equilibrium study of complexes of Mn^2+^ with pyrazine, quinoxaline or phenazine and its carbonate species at 298 K. For the complexes of manganese(II)–pyrazine, manganese(II)–quinoxaline and manganese(II)–phenazine, the formation constants obtained were log β_110_ = 4.6 ± 0.1, log β_110_ = 5.9 ± 0.1 and log β_110_ = 6.0 ± 0.1, respectively. The formation constants for the carbonated species manganese(II)–carbonate, manganese(II)–pyrazine–carbonate, manganese(II)–quinoxaline–carbonate and manganese(II)–phenazine–carbonate complexes were log β_110_ = 5.1 ± 0.1, log β_110_ = 9.8 ± 0.1, log β_110_ = 11.7 ± 0.1 and log β_110_ = 12.7 ± 0.1, respectively. Finally, the individual calculated electronic spectra and its distribution diagram of these species are also reported. The use of N-donor ligand with π-electron-attracting activity in a manganese(II) complex might increase its interaction with carbonate ions.

## 1. Introduction

Energy generation is based in the combustion of fossil fuels, which causes the artificial emission of CO_2_, promoting severe anthropogenic climate change [1,2] and several other consequences [3,4,5]. CO_2_ in adequate quantities allows the planet to maintain a constant temperature. If this proportion is exceeded, the atmospheric heat dissipation decreases. Additionally, together with methane, they are considered the most important gases which cause greenhouse effects that contribute to global warming [6]. In order to reduce atmospheric CO_2_, new technologies have to be developed [2,7]. In nature, the capture and storage of CO_2_ can occur through the formation of metal carbonates or carbon mineralization [8]; in this process, CO_2_ is captured by inorganic salts containing magnesium or calcium, providing an inorganic carbonate which can usually be used in the construction industry [9,10].

Carbon capture and storage technologies are required to diminish the large quantities of CO_2_ emitted, which have been defined by the Intergovernmental Panel on Climate Change [9,10]. On the other hand, CO_2_ that has been captured can be transformed into useful products, only if its utilization promotes a new equilibrium through the generation of an artificial carbon cycle [9,10]. In addition to CO_2_ sequestration technologies, carbonate-based sequestration ideas, concepts and technologies have been barely explored, and thus, they are described as an immature technology [10]. The conversion to carbonate presents several advantages, for example it can be stored as a solid, which confines the environmental impacts to a small spatial volume [10,11]. The development of a carbonate sequestration technology is a complement of CO_2_ sequestration technology, which might assure its long-term viability [10,11]. 

On the other hand, the interactions of Mn^2+^ with carbonates have been reported in natural [12] and environmental [13] systems. These studies have been mainly focused on the solubility of rhodochrosite [14] and coprecipitation of Mn^2+^ with calcite, aragonite or the adsorption or exchange of Mn^2+^ on the surfaces of a calcium carbonate mineral [12] or goethite [15]. Several studies on the adsorption of Mn^2+^ in calcite indicated that could participate in an equilibrium exchange with calcium carbonate, but its mechanism remains elusive [12,14]. In this work, in order to explore the behavior of the Mn^2+^ ion with several ligands and its interaction with carbonates, we report an equilibrium study to obtain the formation constant of Mn^2+^ complexes with the ligands pyrazine (pz), quinoxaline (qx) or phenazine (fz) and its carbonate complexes. This information is compared with manganese(II) carbonate. Pyrazine is natural compound used as a flavoring in foods [16,17], and quinoxaline has a wide range of biological activities such as antibacterial, antitubercular, antiviral, antifungal, antiprotozoan, antiparasitic and as a non-steroidal anti-inflammatory drug [18]. Finally, phenazine is produced by a diverse range of bacteria [19] and is used with multiple research purposes, which range from materials to biological sciences [20,21]. As far as we know, the formation constants of Mn^2+^ with the ligands pyrazine, quinoxaline or phenazine and its interaction with carbonates have not been reported before. Nevertheless, the stability constant of Mn(II) with bicarbonate and its dependence on temperature has been reported before [22].

## 2. Results and Discussion 

### 2.1. Materials 

The methanol HPLC was the solvent used for the experiments, considering its effect on carbonate equilibrium and its role in preventing the formation of a solid hydrate [23]. The proximity of the donor numbers between methanol (19) and water (18) allows us to expect an analogy between the nucleophilic effects in both solution systems [24]. Solvation spheres of the Mn(II) ion with methanol and water are similar [25]. In order to prevent an early precipitation of the complexes, the use of ionic strength was not used; this allows the use of a wide concentration range of ligands (either pyrazine, quinoxaline, phenazine or carbonate). The formation constants in this study should not be taken as stability constants and must be used only to compare systems measured in similar conditions.

### 2.2. Formation Constants of the Manganese(II) Complexes with Pyrazine, Quinoxaline or PhenaZine and Its Carbonate Complexes

As far as we know, there are no reports concerning the formation constants of the manganese(II) complexes with pyrazine, quinoxaline or phenazine in methanol or any other solvent. Nevertheless, a few manganese(II) complexes with pyrazine- [26,27,28,29], quinoxaline- [30,31,32] or phenazine- [33,34] based ligands have been reported before. The electronic spectra of the methanol solutions of manganese(II)–pyrazine complexes are shown in Appendix A, those for the manganese(II)–quinoxaline complex are shown in Appendix A and those for the manganese(II)–phenazine complex are shown in Appendix A. 

Several peaks appear at low ligand concentrations. In all cases, as the concentration increases, a hyperchromic effect is observed. For the manganese(II)–pyrazine complex the peaks appeared at 214 and 260 nm, for the manganese(II)–quinoxaline complex they appeared at 205, 232 and 313 nm, and finally for the manganese(II)–phenazine complex, the peaks appeared at 205, 247 and 361 nm. The obtained values of the formation constants correspond to equilibrium between Mn^2+^ and each ligand. The formation constants log β_jkl_ were obtained by processing the experimental spectra with the software HypSpec [35]. The process consists of a correlation of the spectra of two experiments (each at unique Mn^2+^ concentrations using two different ranges of concentrations of the ligand).

A proposal of the possible colored species generated in solution, an equilibrium model and a value of its formation constant should be initially suggested. For all systems, only two colored species including Mn^2+^ were found. By considering that these ligands usually behave as monodentate, the formation constants were achieved using the next model, where L is pz (pyrazine), qx (quinoxaline) or fz (phenazine):Mn^2+^ + pz ⇌ [Mn(pz)]^2+^  log β_110_(1)
Mn^2+^ + qx ⇌ [Mn(qx)]^2+^  log β_110_(2)
Mn^2+^ + fz ⇌ [Mn(fz)]^2+^   log β_110_(3)

In Table 1, the logarithmic values of the obtained formation constants and a summary of the experimental parameters employed in this study are reported. The calculated electronic spectra of the complexes: manganese(II)–pyrazine, manganese(II)–quinoxaline and manganese(II)–phenazine in methanol, are presented in Figure 1, Figure 2 and Figure 3, respectively.

The calculated electronic spectra of the [Mn(pz)]^2+^ show absorption peaks at 216 nm with a ε = 7100 L mol^−1^ cm^−1^, at 254 nm with ε = 3371 L mol^−1^ cm^−1^ and at 311 nm with ε = 629 L mol^−1^ cm^−1^. On the other hand, the molar absorbance of pyrazine presents a molar absorbance at 261 nm with a ε = 5060 L mol^−1^ cm^−1^ and at 311 with ε = 641 L mol^−1^ cm^−1^.

For the complex [Mn(qx)]^2+^, the absorption peaks are observed at 209 nm with a ε = 23,008 L mol^−1^ cm^−1^, 232 nm with a ε = 18,080 L mol^−1^ cm^−1^ and 315 nm with a ε = 6129 L mol^−1^ cm^−1^. Additionally, the quinoxaline presents a molar absorbance at 202 nm with ε = 14,419 L mol^−1^ cm^−1^, at 233 nm with ε = 24,913 L mol^−1^ cm^−1^ and at 315 nm with a ε = 6019 L mol^−1^ cm^−1^. According to Figure 2, several signals of the quinoxaline change with the coordination of manganese (II), except the peak at 315 nm.

Finally, for the complex [Mn(fz)]^2+^ the absorption peaks are observed at 207 nm with a ε = 51,269 L mol^−1^ cm^−1^, 242 nm with a ε = 58,831 L mol^−1^ cm^−1^ and 362 nm with a ε = 16,208 L mol^−1^ cm^−1^. Additionally, the phenazine presents a molar absorbance at 208 nm with ε = 21,447 L mol^−1^ cm^−1^, at 248 nm with ε = 113,385 L mol^−1^ cm^−1^ and at 363 nm with a ε = 14,736 L mol^−1^ cm^−1^. According to Figure 3, except the peak at 363 nm, several signals of the phenazine change with the coordination of manganese (II), for example the signal at about 207 nm suffers an hyperchromic change, the signal at 248 nm suffers an hypochromic change and the peak changes from sharp to broad. There is a great difference in the absorption of manganese(II) complexes when pyrazine, quinoxaline or phenazine are used, in which the manganese(II)–phenazine generates the complexes with a higher molar absorption. This is probably related to the increase in aromatic character of the ligand. On the other hand, the peaks observed at 216 or 260 nm for the [Mn(pz)]^2+^, at 209 or 232 nm for the [Mn(qx)]^2+^ and 207 or 242 nm for the [Mn(fz)]^2+^, can be assigned to an intra-ligand absorption π → π* [36,37]. Additionally, the peaks observed at 311 nm for [Mn(pz)]^2+^, 315 nm for [Mn(qx)]^2+^ and 363 nm for [Mn(fz)]^2+^ can be assigned to n → π* absorptions [38,39,40].

In this study, the bis, tris or binuclear manganese(II) complexes with pyrazine, quinoxaline or phenazine were not found. Nevertheless, in other studies with manganese(II) complexes using pyrazine as a ligand, the *bis*, *tris* or binuclear complexes were also not obtained [27,28]. Possibly, a preventing effect on the formation of these complexes can be related to the coordination geometry of the manganese, in which the manganese(II) is ring-out-of-plane of the ligand. This phenomenon has been observed in manganese–pyrazine complexes [41]. In order to evaluate this theory, a crystallographic study of these complexes should be carried out.

To study the formation of the manganese(II)–ligand–carbonate complexes (where the ligand is pyrazine, quinoxaline or phenazine), an earlier experiment on the formation of manganese(II)–carbonate has to be conducted. In order to do this, we prepared, in methanol, a molar solution of manganese(II) nitrate which was combined with solutions of sodium carbonate, and the collected spectra were obtained by gradually increasing the concentration of carbonate in several solutions. The electronic spectra of the methanol solutions of manganese(II)–carbonate complexes are shown in Appendix A, and in these solutions, a peak at 215 nm appears at a low carbonate concentration, and as the concentration increases, a hyperchromic effect is observed.

The formation constants of the manganese(II)–carbonate complexes were obtained using the same process as described above, using the model:(4)$${\mathrm{Mn}}^{2+}\;+\;{CO}_{3}^{2-}\;⇌\;{\left[\mathrm{Mn}\right]}^{2+}{CO}_{3}^{2-}\;\;\log\;\beta_{101}$$

In Table 2, the logarithmic value of [Mn]^2+^${CO}_{3}^{2-}$ is presented, and the calculated electronic spectra of the manganese(II)–carbonate in methanol are presented in Figure 4. The calculated electronic spectra of [Mn]^2+^${CO}_{3}^{2-}$ show an absorption peak at 217 nm with a ε = 11676 L mol^−1^ cm^−1^.

In order to obtain the ternary complexes between manganese(II)–ligand (either pyrazine, quinoxaline or phenazine) and carbonate, we prepared a molar solution in methanol of manganese(II) nitrate and combined it with a solution of about two molar equivalents of the ligand. As can be seen below, this generates a solution of [Mn(ligand)]^2+^ at the highest formation percentage relative to manganese(II) (where the ligand is pyrazine, quinoxaline or phenazine). Later, a solution of sodium carbonate was added. The spectra of the ternary system were obtained by gradually increasing carbonate concentration in the solutions. 

The electronic spectra of the methanol solutions of manganese(II)–pyrazine–carbonate complexes are shown in Appendix A, those for the manganese(II)–quinoxaline–carbonate are shown in Appendix A and those for the manganese(II)–phenazine–carbonate are shown in Appendix A. For these solutions, several peaks appear at a low carbonate concentration. In all cases, as the concentration increases, a hyperchromic effect is observed. For the manganese(II)–pyrazine–carbonate, the peaks appeared at 207, 260 and 311 nm; for the manganese(II)–quinoxaline–carbonate, they appeared at 211, 235 and 317 nm; finally, for the manganese(II)–phenazine–carbonate complex, the peaks appeared at 206, 249 and 361 nm.

The formation constants log β_jkl_ were obtained by processing the experimental spectra as described above, and the evaluation of the formation constants was achieved using the model: (5)$${\mathrm{Mn}}^{2+}\;+\;\mathrm{pz}\;+\;{CO}_{3}^{2-}\;⇌\;{\left[\mathrm{Mn}(\mathrm{pz})\right]}^{2+}{CO}_{3}^{2-}\;\;\log\;\beta_{111}$$
(6)$${\mathrm{Mn}}^{2+}\;+\;\mathrm{qx}\;+\;{CO}_{3}^{2-}\;⇌\;{\left[\mathrm{Mn}(\mathrm{qx})\right]}^{2+}{CO}_{3}^{2-}\;\;\log\;\beta_{111}$$
(7)$${\mathrm{Mn}}^{2+}\;+\;\mathrm{fz}\;+\;{CO}_{3}^{2-}\;⇌\;{\left[\mathrm{Mn}(\mathrm{fz})\right]}^{2+}{CO}_{3}^{2-}\;\;\log\;\beta_{111}$$

The logarithmic values of the formation constants for the species [Mn]^2+^${CO}_{3}^{2-}$, [Mn(pz)]^2+^${CO}_{3}^{2-}$, [Mn(qx)]^2+^  ${CO}_{3}^{2-}$ and [Mn(fz)]^2+^${CO}_{3}^{2-}$ are presented in Table 2. It can be observed that [Mn(fz)]^2+^${CO}_{3}^{2-}$ has a higher value to stabilize carbonate, and this behavior is probably related to the increase in the aromatic character of the ligand which might potentialize its interaction with the carbonate ion. The calculated electronic spectra of [Mn(pz)]^2+^${CO}_{3}^{2-}$, [Mn(qx)]^2+^${CO}_{3}^{2-}$ and [Mn(fz)]^2+^${CO}_{3}^{2-}$ are presented in Figure 5, Figure 6 and Figure 7, respectively.

The calculated electronic spectra for [Mn(pz)]^2+^${CO}_{3}^{2-}$ display three absorption peaks, at 209 nm with ε = 15,240 L mol^−1^ cm^−1^, at 261 nm with ε = 14,495 L mol^−1^ cm^−1^ and 308 nm with ε = 2746 L mol^−1^ cm^−1^. The complex [Mn(qx)]^2+^${CO}_{3}^{2-}$ exhibits three absorption peaks at 206 nm with ε = 41,403 L mol^−1^ cm^−1^, 233 nm with ε = 43,797 L mol^−1^ cm^−1^ and 314 nm with ε = 10,877 L mol^−1^ cm^−1^. 

Finally, the complex [Mn(fz)]^2+^${CO}_{3}^{2-}$ also displays three absorption peaks at 207 nm with ε = 111,790 L mol^−1^ cm^−1^ and at 247 nm with ε = 232,940 L mol^−1^ cm^−1^ and 361 nm with ε = 44,869 L mol^−1^ cm^−1^. The spectrum of the complexes with carbonate maintains its maximum wavelength absorption, resembling the spectra of non-carbonated complexes, but its molar absorbance is augmented.

The Mn^2+^ has a strong affinity for carbonate ions [42]. When the formation constants of the carbonated complexes are compared with Mn^2+^ complexes with pyrazine, quinoxaline or phenazine, the ligand might help to promote this interaction. These ligands are electron-withdrawing n-heterocycles which can be used as π-electron-attracting ligands [43]. These characteristics, in combination with Mn^2+^ properties, could promote a higher affinity for the carbonate ion, in particular, [Mn(fz)]^2+^. Possibly, the phenazine ligand in combination with Mn^2+^ might have a higher π-character and therefore an increased affinity for carbonate ions.

### 2.3. Distribution Curves of the Manganese(II) Complexes with Pyrazine, Quinoxaline and Phenazine and Its Carbonated Complexes

The speciation diagrams of the manganese(II)–pyrazine, manganese(II)–quinoxaline and manganese(II)–phenazine systems are shown in Figure 8a–c, respectively. A solution with equimolar concentrations of manganese(II) and pyrazine (at 0.000348 M or 0.000696 M, for low- and high-concentration experiments, respectively), roughly yields 77% of the manganese(II)–pyrazine complex and about 23% of free manganese. Yet, two molar equivalents of pyrazine per manganese(II) generates about 94% of manganese(II)–pyrazine complex and 6% of manganese(II).

According to Figure 8b, solutions with equimolar concentrations of manganese(II) and quinoxaline (at 0.00007968 M or 0.00015936 M, for low- and high-concentration experiments, respectively), generate about 88% of the manganese(II)–quinoxaline complex and 12% of manganese(II). On the other hand, two molar equivalents of quinoxaline per manganese(II) generates about 97% of the manganese(II)–quinoxaline complex and 3% of manganese(II). Finally, in Figure 8c it can be observed that solutions with equimolar concentrations of manganese(II) and phenazine (at 0.0000175 M or 0.0000351 M, for low- and high-concentration experiments, respectively), generate about 79% of the manganese(II)–phenazine complex and 21% of manganese(II). Additionally, two molar equivalents of phenazine per manganese(II) generates about 95% of the manganese(II)–phenazine complex and 5% of manganese(II).

Figure 9a–d display the speciation diagrams of the manganese(II)–carbonate, manganese(II)–pyrazine–carbonate, manganese(II)–quinoxaline–carbonate and manganese(II)–phenazine–carbonate, respectively.

An equimolar mixture of manganese(II) and sodium carbonate yields about 86% MnCO_3_ and 14% of manganese(II). Additionally, if two molar equivalents of sodium carbonate are added, a solution of a molar equivalent of manganese (II), a mixture of 2% of manganese(II) and 98% of manganese(II)–carbonate are generated. On the other hand, an equimolar mixture of manganese(II), sodium carbonate and two molar equivalents of pyrazine yields a mixture of 71% of [Mn(pz)]^2+^CO_3_^2−^, 18% of [Mn(pz)]^2+^, 8.6% of (Mn)^2+^(CO_3_)^2−^ and 2.2% (Mn)^2+^. Additionally, if the composition of the mixture changes to a molar equivalent of manganese(II), two molar equivalents of pyrazine and two molar equivalents of sodium carbonate, the composition of the solution changes to 85% of [Mn(pz)]^2+^CO_3_^2−^, 4.5% of [Mn(pz)]^2+^, 10 % of (Mn)^2+^(CO_3_)^2−^ and 0.5% of Mn^2+^. 

However, an equimolar mixture of manganese(II), sodium carbonate and two molar equivalents of quinoxaline yields a mixture of 78% of [Mn(qx)]^2+^${CO}_{3}^{2-}$, 20% of [Mn(qx)]^2+^, 0.75% of (Mn)^2+^(CO_3_)^2−^ and 0.87%(Mn)^2+^. Additionally, if the composition of the mixture changes to a molar equivalent of manganese(II), two molar equivalents of quinoxaline and two molar equivalents of sodium carbonate, the composition of the solution changes to 93.9% of [Mn(qx)]^2+^${CO}_{3}^{2-}$, 5 % of [Mn(qx)]^2+^, 0.9% of (Mn)^2+^(CO_3_)^2−^ and 0.2% of Mn^2+^. Finally, a mixture of a equimolar mixture of manganese(II), sodium carbonate and two molar equivalents of phenazine yields a mixture of 89% of [Mn(fz)]^2+^${CO}_{3}^{2-}$, 10.26% of [Mn(fz)]^2+^, 0.14% of (Mn)^2+^(CO_3_)^2−^ and 0.57%(Mn)^2+^. If the composition of the mixture changes to a molar equivalent of manganese(II), two molar equivalents of phenazine and two molar equivalents of sodium carbonate, the composition of the solution changes to 98.52% of the [Mn(fz)]^2+^${CO}_{3}^{2-}$, 1.11% of [Mn(fz)]^2+^, 0.16 % of (Mn)^2+^(CO_3_)^2−^ and 0.06% of Mn^2+^.

### 2.4. Far- and Mid-Infrared Spectra of the Complexes: [Mn(pz)]^2+^, [Mn(qx)]^2+^, [Mn(fz)]^2+^, [Mn(pz)]^2+^${CO}_{3}^{2-}$, [Mn(qx)]^2+^${CO}_{3}^{2-}$ and [Mn(fz)]^2+^${CO}_{3}^{2-}$

The far- and mid-infrared spectra of the complexes [Mn(pz)]^2+^, [Mn(qx)]^2+^, [Mn(fz)]^2+^, [Mn(pz)]^2+^${CO}_{3}^{2-}$, [Mn(qx)]^2+^${CO}_{3}^{2-}$ and [Mn(fz)]^2+^${CO}_{3}^{2-}$ are shown in Appendix A, respectively. The spectra of the complexes show several similar signals between each complex and the carbonated analogous (their signals are assigned in Table 3). The manganese complexes with pyrazine and quinoxaline and phenazine present signals between 1600 and 1200 cm^−1^ which are associated with ring vibrations of the ligand [44]. These complexes also present a signal at 3330 cm^−1^ assigned to ν(NH); nevertheless, this signal can be shifted to 3290 cm^−1^ for manganese complexes with phenazine [41,45]. In [Mn(pz)]^2+^ and [Mn(pz)]^2+^ ${CO}_{3}^{2-}$, the signal at 450 cm^−1^ (Appendix A) has been assigned to a particular vibrational mode of the coordination of the manganese to nitrogen in the pyrazine ring [41,46]. In compounds, the signal at 250 cm^−1^ is assigned to a vibration mode ν(Mn–N) [41]. Finally, the bands at 400 and 300 cm^−1^ (Appendix A) can be assigned to a vibration mode ν(Mn–O) (either oxygen possibly from carbonate or the methanol used as solvent, which depends on each sample) [41].

## 3. Materials and Methods

### 3.1. Materials, Physical Measurements and Methods

For the spectral measurements methanol HPLC-grade (Karal, Mexico) was used as a solvent; the manganese(II) nitrate tetrahydrate, Mn(NO_3_)_2_·4H_2_O (Sigma-Aldrich, USA), pyrazine(Sigma-Aldrich, USA), quinoxaline (Sigma-Aldrich, India), phenazine (Sigma-Aldrich, Germany) and sodium carbonate Na_2_CO_3_ (Karal, Mexico) were analytical grade and used without further purification. The spectral measurements were performed at 298 K in a quartz cell with a 1 cm path length and 3 mL volume. We used a Shimadzu UV–vis-1800 spectroscopy system equipped with a Thermo Scientific thermostat system TPS-1500W. The spectrophotometric data were fitted in the program HypSpec [47,48] and the distribution diagrams of species were calculated in the software Hyperquad Simulation and Speciation (HySS2009, Leeds, UK) [49]. 

In a typical spectral measurement of the manganese(II)–ligand complexes, stock solutions of manganese (II) and ligand (either pyrazine, quinoxaline or phenazine) were prepared and diluted to obtain a solution behaving according the Beer–Lambert law. The final concentration of the Mn(II) ion is kept as constant, and the concentration of the ligand (either pyrazine, quinoxaline or phenazine) is varied within a range. This process is repeated twice, and each experiment is carried out at a different concentration of metal ions and at two different ranges of ligand concentration.

On the other hand, to decide on the concentrations of manganese(II) ions and ligands to generate the species of manganese(II)–ligand–carbonate, the concentrations chosen were such that they would ensure the highest amounts of the manganese(II)–ligand, according to its speciation diagrams shown above. At these concentrations of the manganese(II) ion and ligand, to generate each carbonated species we added a range of concentrations of carbonate ions. In typical manganese(II)–ligand–carbonate experiments, the final concentration of the manganese ion and the ligand (either pyrazine, quinoxaline or phenazine) were maintained at a constant and the concentration of the carbonate ion varied within a range. In order to obtain a spectral measurement in solution and to avoid turbidity (haze) or precipitation caused by the increase in carbonate concentration, several concentration ranges of carbonate were tested. Here, we only reported the concentration ranges that allowed us to obtain a spectral measurement of the interaction between the complexes of Mn(II)–ligand (either pyrazine, quinoxaline or phenazine) and carbonate ion in methanol solution without haze or precipitation. For the determination of the formation constants, in all the experiments the spectral region analyzed was from 200 to 420 nm.

### 3.2. Equilibrium Studies of Manganese (II) with Pyrazine, Quinoxaline or Phenazine

For the manganese(II)–pyrazine system, we prepared two different stock solutions of pyrazine (350 and 700 mM) and Mn(NO_3_)_2_·4H_2_O (348 and 696 mM). For the experiments, the final Mn^2+^ concentration remained constant at 348 and 696 μM. For each experiment, the concentrations of pyrazine varied from 35 to 671 μM and 71 to 1412 μM, respectively. The collected spectra of the solutions are shown in Appendix A, and a total of 40 spectra were used for the refinement. For the manganese(II)–quinoxaline system, were prepared two different stock solutions of quinoxaline (61.5 and 153.85 mM) and Mn(NO_3_)_2_·4H_2_O (79.68 and 159.36 mM). For the experiments, the final Mn^2+^ concentration remained constant at 79.68 and 159.36 μM. For each experiment the concentrations of quinoxaline were varied from 6.14 to 167.81 μM and 15.36 to 245.85 μM, respectively. A total of 36 spectra were used for the refinement, and the collected spectra of the solutions are shown in Appendix A.

For the manganese(II)–phenazine system, we prepared two different stock solutions of phenazine (17.76 and 35.51 mM) and Mn(NO_3_)_2_·4H_2_O (17.53 and 35.06 mM). For the experiments, the final Mn^2+^ concentration remained constant at 17.53 and 35.06 μM, and for each experiment the concentrations of phenazine varied from 1.78 to 33.74 μM and 3.55 to 71.03 μM, respectively. A total of 40 spectra were used for the refinement, and the collected spectra of the solutions are shown in Appendix A.

### 3.3. Equilibrium Studies of Manganese (II) with ${CO}_{3}^{2-}$ and Pyrazine, Quinoxaline or Phenazine

In order to generate the manganese(II)–carbonate species, we used a stock solution of Mn(NO_3_)_2_·4H_2_O (334.7 mM) and sodium carbonate (133.8 mM). For these experiments, the final Mn^2+^ concentration was set constant at 334.7 μM and the carbonate concentration was varied from 133.8 to 602.5 μM. For the refining process, a total of 10 spectra were used, which are shown in Appendix A.

For the manganese(II)–pyrazine–carbonate species, two stock solutions of Mn(NO_3_)_2_·4H_2_O (159.4 and 318.7 mM), pyrazine (348.9 and 697.8 mM), and sodium carbonate (300 and 600 mM) were prepared. For these experiments, the final Mn^2+^ concentration was set constant at 159.2 and 334 μM, and pyrazine, at 349.44 and 700 μM, respectively. The carbonate concentrations varied from 30.18 to 301.88 and from 66.94 to 669.4 μM, respectively. A total of 20 spectra were used for the refining process, and the collected spectra of the solutions are shown in Appendix A.

On the other hand, for the complex manganese(II)–quinoxaline–carbonate, two stock solutions of Mn(NO_3_)_2_·4H_2_O (39.84 and 79.68 mM), quinoxaline (61.5 and 123 mM) and sodium carbonate (80 and 160 mM) were prepared. For these experiments, the final Mn^2+^ concentration was set constant at 31.84 and 80 μM and quinoxaline was set at 61.44 and 122.88 μM, respectively. The carbonate concentration varied from 7.54 to 75.4 μM and 16 to 160 μM, respectively. For the refining process, a total of 20 spectra were used and are shown in Appendix A.

Finally, for the manganese(II)–phenazine–carbonate system, two stock solutions of Mn(NO_3_)_2_·4H_2_O (9.56 and 19.12 mM), phenazine (177.57 and 355.14 mM) and sodium carbonate (94.34 and 188.68 mM) were prepared. For these experiments, the final Mn^2+^ concentration was set constant at 9.55 and 17.6 μM and phenazine was set at 17.76 mM and 35.52 μM, respectively. The carbonate concentration varied from 0.94 to 8.49 μM and 7.54 to 37.7 μM, respectively. A total of 19 spectra were used for the refining process, and the collected spectra of the solutions are shown in Appendix A.

### 3.4. Synthesis of the Complexes: [Mn(pz)]^2+^, [Mn(qx)]^2+^, [Mn(fz)]^2+^, [Mn(pz)]^2+^${CO}_{3}^{2-}$, [Mn(qx)]^2+^${CO}_{3}^{2-}$, [Mn(fz)]^2+^${CO}_{3}^{2-}$ and Its Far- and Mid-Infrared Spectrum

The Mn^2+^ complexes were prepared in methanol HPLC (100 mL) by combining, in solution, an amount of the manganese(II) nitrate tetrahydrate and the ligand used. In order to prepare the [Mn(ligand)]^2+^, an equimolar solution of manganese(II) nitrate nonahydrate (250 mM) and the ligand (250 mM) were mixed in methanol. Additionally, to prepare the [Mn(ligand)]^2+^${CO}_{3}^{2-}$, an equimolar solution of manganese(II) nitrate nonahydrate (25 mM), ligand (25 mM) and sodium carbonate (25 mM) were mixed in methanol (where the ligand is either pyrazine, quinoxaline or phenazine). The solutions remained at 4 °C until precipitation. Once the complexes formed, the solutions were filtered to collect the product. If solid samples are redissolved and their UV–vis spectrum is obtained, the signals are similar to the UV–vis spectrum calculated for the complexes. The far- and mid-infrared spectra for the different manganese complexes were obtained using an HATR system in a Perkin-Elmer Frontier FTIR/FIR spectrometer in ranges from 700 to 50 and 1400 to 400 cm^−1^, respectively.

## 4. Conclusions

A carbonate exchange or storing agent using manganese (II) has several applications, which range from the stabilization of artificial carbon sinks to anode material in lithium batteries. The study of manganese(II)–carbonate complexes is a potential research area. The use of a suitable ligand in a manganese(II) complex might potentiate its interaction with carbonates. In our study, the formation constants obtained for the carbonated complexes were higher, in particular for [Mn(fz)]^2+^${CO}_{3}^{2-}$. The use of an N-donor ligand with π-electron-attracting activity in manganese(II) complexes might increase its interaction with carbonate ions. The possible coordination or electrostatic attraction of the carbonate ion to the manganese complex should be confirmed using a further crystallographic study. This might help us to understand the role of manganese geometry in generating only mononuclear complexes and its carbonate interactions.

## Figures and Tables

**Figure 1 molecules-27-01648-f001:**
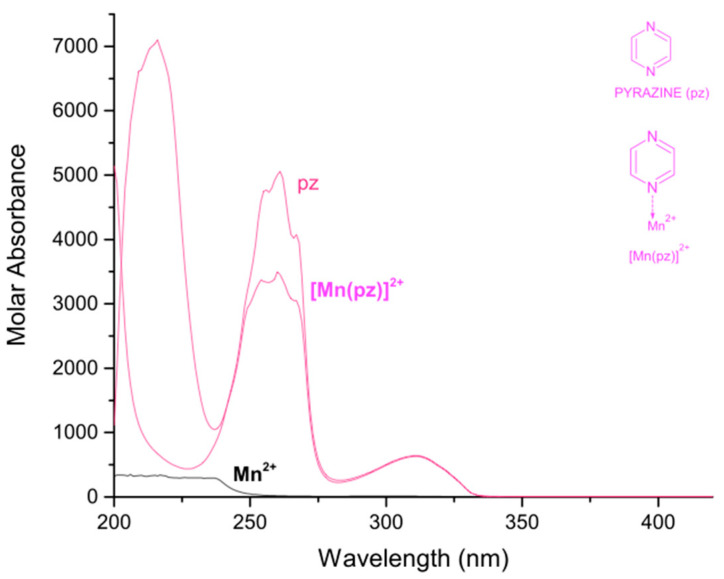
Calculated electronic spectrum of manganese(II)–pyrazine in methanol: pyrazine (pz), Mn^2+^ and [Mn(pz)]^2+^.

**Figure 2 molecules-27-01648-f002:**
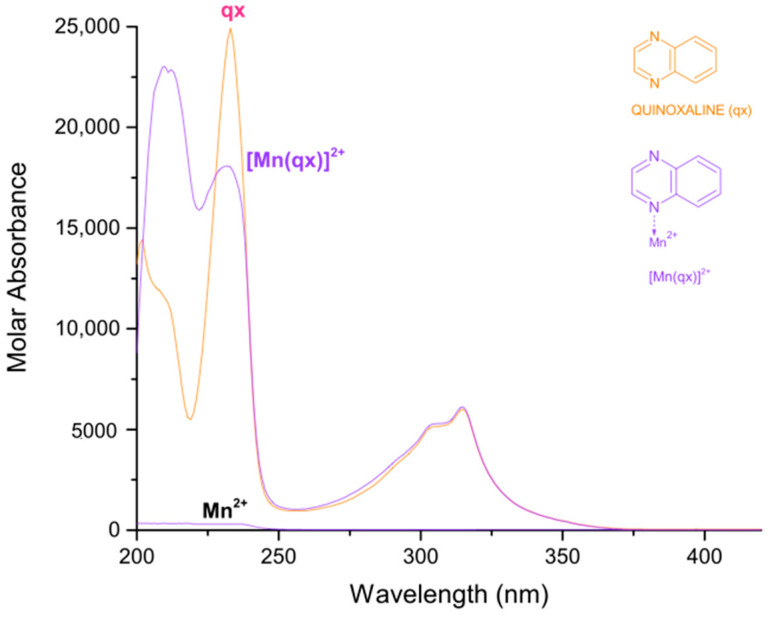
Calculated electronic spectrum of manganese(II)–quinoxaline in methanol:quinoxaline (qx), Mn^2+^ and [Mn(qx)]^2+^.

**Figure 3 molecules-27-01648-f003:**
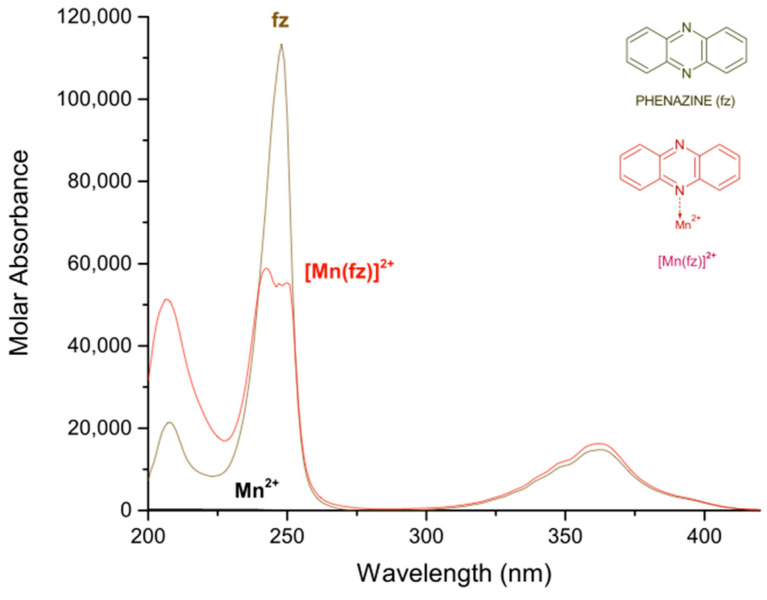
Calculated electronic spectrum of manganese(II)–phenazine in methanol: phenazine (fz), Mn^2+^ and [Mn(fz)]^2+^.

**Figure 4 molecules-27-01648-f004:**
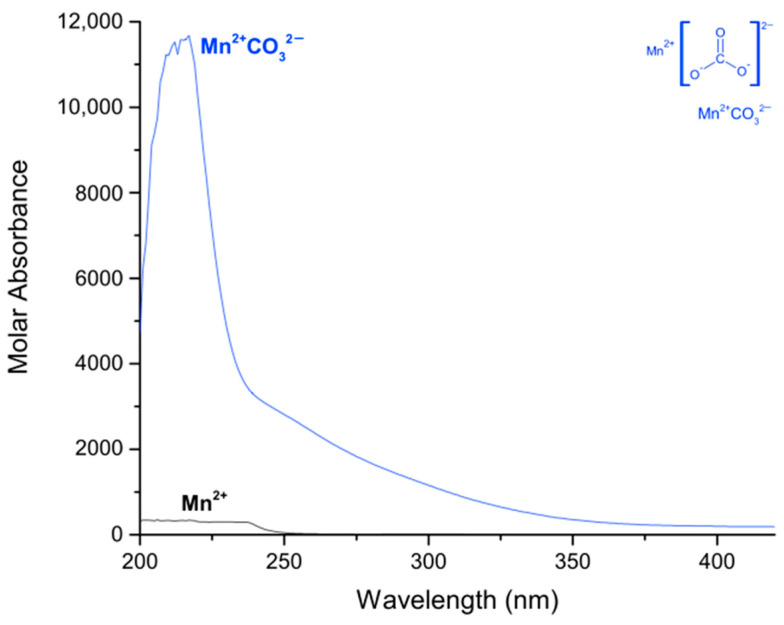
Calculated electronic spectrum of manganese(II)–carbonate in methanol: Mn^2+^ and [Mn(CO_3_)]^2+^.

**Figure 5 molecules-27-01648-f005:**
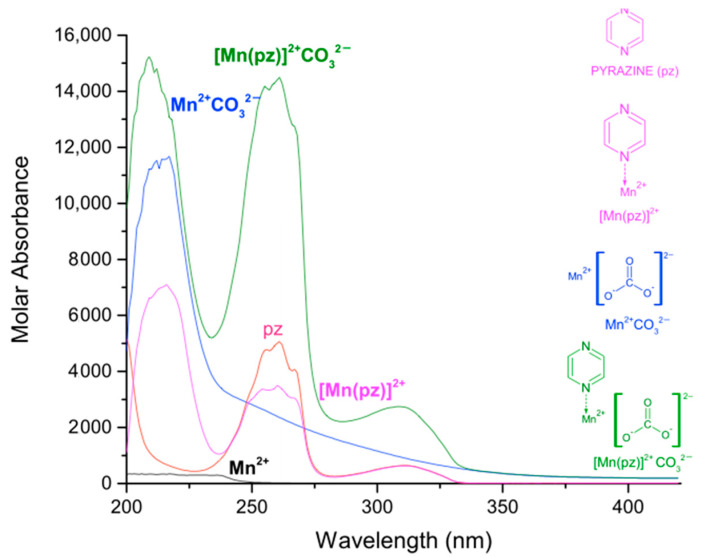
Calculated electronic spectrum of manganese(II)–pyrazine–carbonate in methanol: Mn^2+^, pz, Mn^2+^${CO}_{3}^{2-}$, [Mn(pz)]^2+^ and [Mn(pz)]^2+^${CO}_{3}^{2-}$.

**Figure 6 molecules-27-01648-f006:**
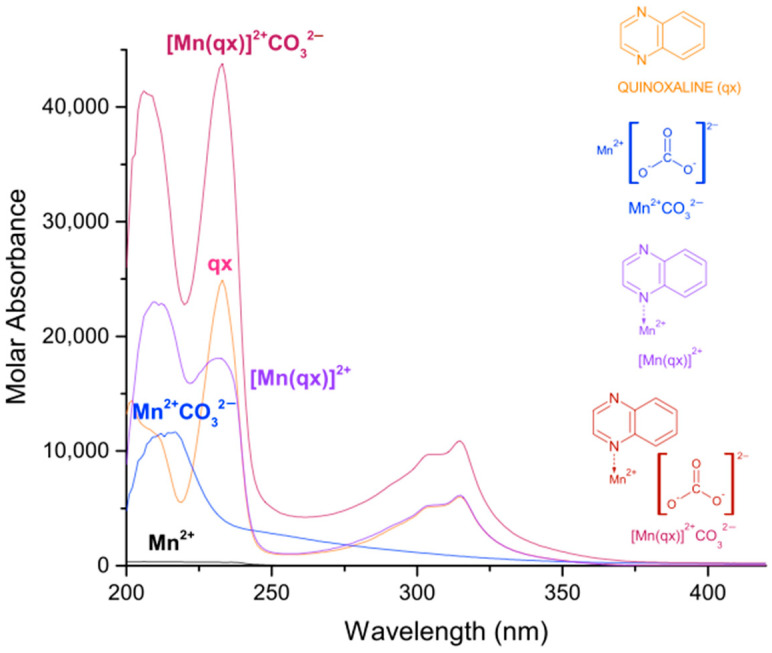
Calculated electronic spectrum of manganese(II)–quinoxaline–carbonate in methanol: Mn^2+^, [Mn(qx)]^2+^, Mn^2+^${CO}_{3}^{2-}$ and [Mn(qx)]^2+^${CO}_{3}^{2-}$.

**Figure 7 molecules-27-01648-f007:**
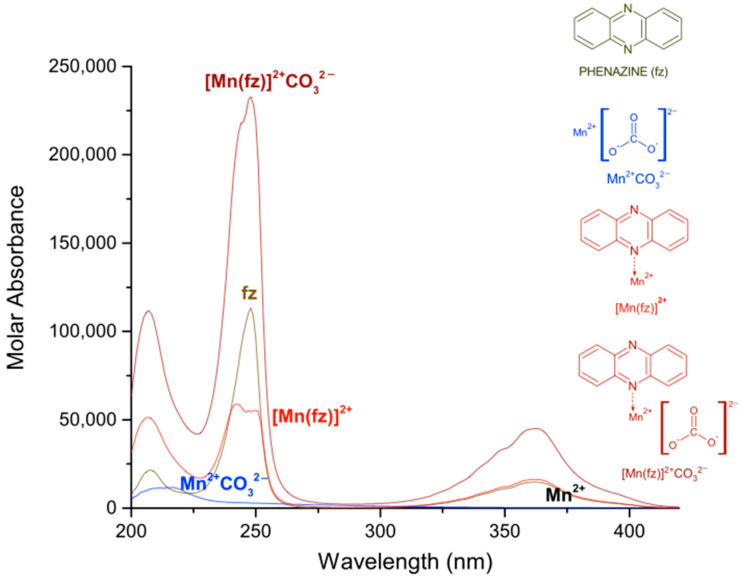
Calculated electronic spectrum of manganese(II)–phenazine–carbonate in methanol: Mn^2+^, [Mn(fz)]^2+^, Mn^2+^${CO}_{3}^{2-}$ and [Mn(fz)]^2+^${CO}_{3}^{2-}$.

**Figure 8 molecules-27-01648-f008:**
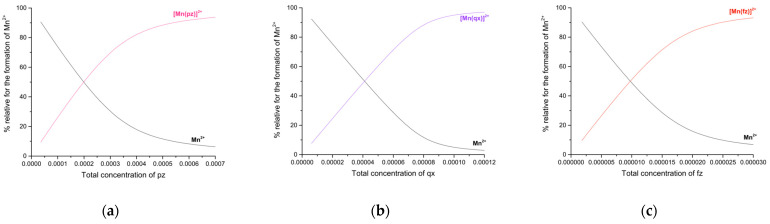
(**a**) Formation curves of the manganese(II)–pyrazine complexes in methanol. [Mn]^2+^ = 348 µM and pyrazine range from 35 to 706 µM. (**b**) Formation curves of the manganese(II)–quinoxaline complexes in methanol. [Mn]^2+^ = 80 µM and quinoxaline range from 6 to 123 µM. (**c**) Formation curves of the manganese(II)–phenazine complexes in methanol. [Mn]^2+^ = 17.5 µM and phenazine range from 1.78 to 35.5 µM.

**Figure 9 molecules-27-01648-f009:**
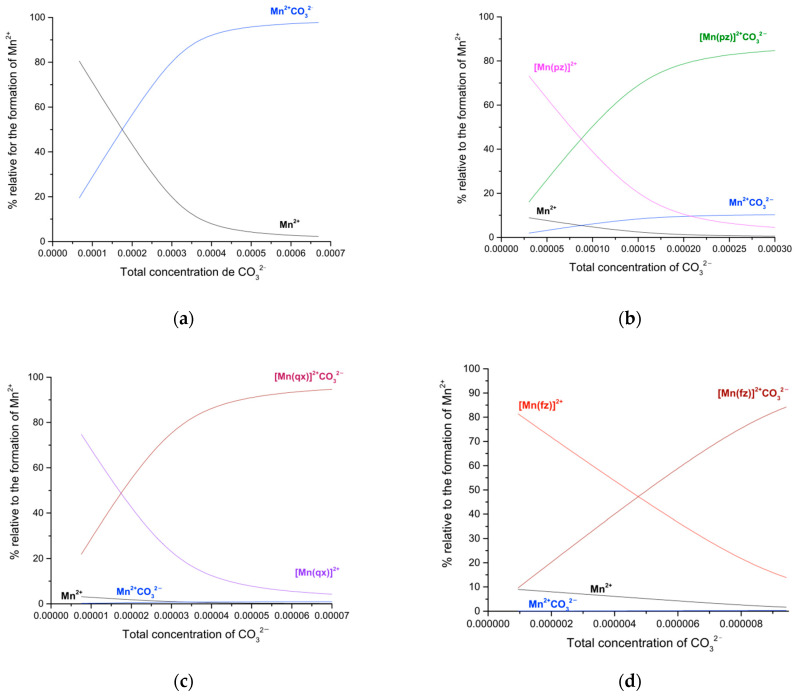
(**a**) Formation curves of the manganese(II)–carbonate complexes in methanol. [Mn]^2+^ = 335 µM and sodium carbonate range from 66.9 to 669 µM. (**b**) Formation curves of the manganese(II)–pyrazine–carbonate complexes in methanol. [Mn]^2+^ = 159 µM, [pz] = 349 µM and Sodium carbonate range from 30.2 to 302 µM. (**c**) Formation curves of the manganese(II)–quinoxaline–carbonate complexes in methanol. [Mn]^2+^ = 31.8 µM, [qx] = 61.4 µM and Sodium carbonate range from 7.54 to 75.4 µM. (**d**) Formation curves of the manganese(II)–phenazine–carbonate complexes in methanol. [Mn]^2+^ = 9.6 µM, [fz] = 17.8 µM and Sodium carbonate range from 0.9 to 9.4 µM.

**Table 1 molecules-27-01648-t001:** Summary of experimental parameters for the systems Mn^2+^ with pyrazine (pz), quinoxaline (qx) and phenazine (fz), in methanol.

Solution Composition		[T_L_] Range from 35 to 671 and 71 to 1412 µmol L^−1^ [T_M_] Constant at 348 and 696 µmol L^−1^	
		Ionic strength, electrolyte	Not used
		pH range	Not used
Experimental method		Spectrophotometric titration	
Temperature		298 K	
Total number of data points		Mn complexation: 40 solution spectra	
Method of calculation		HypSpec	
Species	Equilibrium	Log β	σ
[Mn(pz)]^2+^	Mn^2+^ + pz ⇌ [Mn(pz)]^2+^	log β_110_ = 4.6 ± 0.1	0.0036
Solution composition		[T_L_] range from 6.14 to 167.81 and 15.36 to 245.85 µmol L^−1^ [T_M_] constant at 79.68 and 159.36 µmol L^−1^	
		Ionic strength, electrolyte	Not used
		pH range	Not used
Experimental method		Spectrophotometric titration	
Temperature		298 K	
Total number of data points		Mn complexation: 36 solution spectra	
Method of calculation		HypSpec	
Species	Equilibrium	Log β	σ
[Mn(qx)]^2+^	Mn^2+^ + qx ⇌ [Mn(qx)]^2+^	log β_110_ = 5.9 ± 0.1	0.0278
Solution composition		[T_L_] range from 1.78 to 33.74 and 3.55 to 71.03 µmol L^−1^ [T_M_] constant at 17.53 and 35.06 µmol L^−1^	
		Ionic strength, electrolyte	Not used
		pH range	Not used
Experimental method		Spectrophotometric titration	
Temperature		298 K	
Total number of data points		Mn complexation: 40 solution spectra	
Method of calculation		HypSpec	
Species	Equilibrium	Log β	σ
[Mn(fz)]^2+^	Mn^2+^ + fz ⇌ [Mn(fz)]^2+^	log β_110_ = 6.0 ± 0.1	0.0161

**Table 2 molecules-27-01648-t002:** Summary of experimental parameters for the systems Mn(II)–carbonate and Mn(II)–ligand–carbonate, where the ligand is pyrazine (pz), quinoxaline (qx) or phenazine (fz), in methanol.

Solution Composition		[T_CO3_] Range from 133.8 to 602.5 µmol L^−1^ [T_M_] Constant at 334.7 µmol L^−1^	
		Ionic strength, electrolyte	Not used
		pH range	Not used
Experimental method		Spectrophotometric titration	
Temperature		298 K	
Total number of data points		Mn complexation: 10 solution spectra	
Method of calculation		HypSpec	
Species	Equilibrium	Log β	σ
[Mn]^2+^${CO}_{3}^{2-}$	Mn^2+^ + CO_3_^2−^ ⇌ [Mn]^2+^${CO}_{3}^{2-}$	log β_110_ = 5.1 ± 0.1	0.0817
Solution composition		[T_CO3_] range from 30.18 to 301.88 µmol L^−1^ and 66.94 to 669.4 µmol L^−1^ [T_L_] constant at 349.44 and 700 µmol L^−1^ [T_M_] constant at 159.2 and 334 µmol L^−1^	
		Ionic strength, electrolyte	Not used
		pH range	Not used
Experimental method		Spectrophotometric titration	
Temperature		298 K	
Total number of data points		Mn complexation: 20 solution spectra	
Method of calculation		HypSpec	
Species	Equilibrium	Log β	σ
[Mn(pz)]^2+^${CO}_{3}^{2-}$	Mn^2+^ + pz + ${CO}_{3}^{2-}$ ⇌ [Mn(pz)]^2+^${CO}_{3}^{2-}$	log β_110_ = 9.8 ± 0.1	0.1224
Solution composition		[T_CO3_] range from 7.54 to 75.4 µmol L^−1^ and 16.0 to 160.0 µmol L^−1^ [T_L_] constant at 61.44 and 122.88 µmol L^−1^ [T_M_] constant at 31.84 and 80.0 µmol L^−1^	
		Ionic strength, electrolyte	Not used
		pH range	Not used
Experimental method		Spectrophotometric titration	
Temperature		298 K	
Total number of data points		Mn complexation: 20 solution spectra	
Method of calculation		HypSpec	
Species	Equilibrium	Log β	σ
[Mn(qx)]^2+^${CO}_{3}^{2-}$	Mn^2+^ + qx + ${CO}_{3}^{2-}$ ⇌ [Mn(qx)]^2+^${CO}_{3}^{2-}$	log β_110_ = 11.7 ± 0.1	0.0266
Solution composition		[T_CO3_] range from 0.94 to 8.49 µmol L^−1^ and 7.54 to 37.7 µmol L^−1^ [T_L_] constant at 17.76 and 35.52 µmol L^−1^ [T_M_] constant at 9.55 and 17.6 µmol L^−1^	
		Ionic strength, electrolyte	Not used
		pH range	Not used
Experimental method		Spectrophotometric titration	
Temperature		298 K	
Total number of data points		Mn complexation: 19 solution spectra	
Method of calculation		HypSpec	
Species	Equilibrium	Log β	σ
[Mn(fz)]^2+^${CO}_{3}^{2-}$	Mn^2+^ + fz + ${CO}_{3}^{2-}$ ⇌ [Mn(fz)]^2+^${CO}_{3}^{2-}$	log β_110_ = 12.7 ± 0.1	0.0492

**Table 3 molecules-27-01648-t003:** FIR and NIR spectral data for the complexes [Mn(pz)]^2+^, [Mn(qx)]^2+^, [Mn(fz)]^2+^, [Mn(pz)]^2+^${CO}_{3}^{2-}$, [Mn(qx)]^2+^${CO}_{3}^{2-}$ and [Mn(fz)]^2+^${CO}_{3}^{2-}$.

Complex	Signal of Ring Vibration, cm^−1^	ν(NH), cm^−1^	ν(Mn–N), cm^−1^	ν(Mn–O), cm^−1^
[Mn(pz)]^2+^	1600 and 1200 [44]	3330 cm^−1^ [41,45]	250 cm^−1^ [41]	400 and 300 cm^−1^ [41]
[Mn(qx)]^2+^				
[Mn(fz)]^2+^		3290 cm^−1^ [41,45]		
[Mn(pz)]^2+^${CO}_{3}^{2-}$		3330 cm^−1^ [41,45]		
[Mn(qx)]^2+^${CO}_{3}^{2-}$				
[Mn(fz)]^2+^${CO}_{3}^{2-}$		3290 cm^−1^ [41,45]		

## Data Availability

The data presented in this study are available in Appendix A.

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
