# Peer review of "Stability of Manganese(II)–Pyrazine, –Quinoxaline or –Phenazine Complexes and Their Potential as Carbonate Sequestration Agents"

_molecules, 2022, doi:10.3390/molecules27051648_

Round 1

Reviewer 1 Report

In this work, authors studied the behavior of the Mn2+ ion with several ligands and its interaction con carbonates, and reported an equilibrium study to obtain the formation constant of Mn2+ complexes with the ligands, such as pyrazine (pz), quinoxaline (qx), phenazine 63 (fz) and its carbonate complexes. The formation constants obtained for the carbonated complexes were higher, in particular for the [Mn(fz)]2+CO32-. The use of an N-donor ligand with π-electron attracting activity in manganese(II) complexes, might increase its interaction with carbonate ions.

The related work has not been reported before, and the use of a suitable ligand in a manganese(II) complex might potentiate its interaction with carbonates, which can be as a carbonate exchange or store agent. In my view, this study is interesting and innovative, and only some language polish is needed.

Reviewer 2 Report

This paper submitted by Segoviano-Garfias and co-workers investigated a series of manganese complexes supported by pyrazine, quinoxaline, or phenazine ligands for the use of carbonate agents. They studied the formation constant for their corresponding carbonate. Overall, I think this is a well-written paper, the characterization is complete and the discussion/conclusion makes sense so I support publication with minor revision.

Here are my concerns/suggestions.

1) line 78, methanol HPLC-HPLC grade methanol

2) I suggest the authors add one figure of all complexes' chemdraw in the paper.

3) I don't think it is necessary to show all of those UV-Vis and IR spectra in the manuscript. They can move them to the SI and just make a table in the paper.